# An Intelligent Recommendation for Intelligently Accessible Charging Stations: Electronic Vehicle Charging to Support a Sustainable Smart Tourism City

Pannee Suanpang [1,*], Pitchaya Jamjuntr [2], Phuripoj Kaewyong [1], Chawalin Niamsorn [3] and Kittisak Jermsittiparsert [4,5,6,7,8,*]

1. Faculty of Science & Technology, Suan Dusit University, Bangkok 10300, Thailand
2. Faculty of Engineering, King Mongkut's University of Technology Thonburi, Bangkok 10140, Thailand
3. Faculty of Management Sciences, Suan Dusit University, Bangkok 10300, Thailand
4. Faculty of Education, University of City Island, Famagusta 9945, Cyprus
5. Faculty of Social and Political Sciences, Universitas Muhammadiyah Sinjai, Kabupaten Sinjai 92615, Sulawesi Selatan, Indonesia
6. Faculty of Social and Political Sciences, Universitas Muhammadiyah Makassar, Kota Makassar 90221, Sulawesi Selatan, Indonesia
7. Publication Research Institute and Community Service, Universitas Muhammadiyah Sidenreng Rappang, Sidenreng Rappang Regency 91651, Sulawesi Selatan, Indonesia
8. Sekolah Tinggi Ilmu Administrasi Abdul Haris, Kota Makassar 90000, Sulawesi Selatan, Indonesia
* Correspondence: pannee_sua@dusit.ac.th (P.S.); kittisak.jermsittiparsert@adakent.edu.tr (K.J.)

**Abstract:** The world is entering an era of awareness of the preservation of natural energy sustainability. Therefore, electric vehicles (EVs) have become a popular alternative in today's transportation system as they have zero emissions, save energy, and reduce pollution. One of the most significant problems with EVs is an inadequate charging infrastructure and spatially and temporally uneven charging demands. As such, EV drivers in many large cities frequently struggle to find suitable charging locations. Furthermore, the recent emergence of deep reinforcement learning has shown great promise for improving the charging experience in a variety of ways over the long term. In this paper, a Spatio-Temporal Multi-Agent Reinforcement Learning (STMARL) (Master) framework is proposed for intelligently public-accessible charging stations, taking into account several long-term spatio-temporal parameters. When compared to a random selection recommendation system, the experimental results demonstrate that an STMARL (master) framework has a long-term goal of lowering the overall charging wait time ($CWT$), average charging price ($CP$), and charging failure rate ($CFR$) of EVs.

**Keywords:** electric vehicle; intelligent recommendation system; electronic vehicle charging; smart tourism; destination; smart city

## 1. Introduction

In the digital transformation age, technological changes have created disruptions in all dimensions with the use of advanced information technology, especially artificial intelligence (AI) that has been widely used in every industry worldwide. More recently, electric vehicles (EVs) have been introduced and have received increased attention. This has resulted in eco-friendly cars, which various countries have been encouraging the use of as new modes of transportation, thus bringing many advantages [1], Due to their zero emissions, electric vehicles are considered environmentally benign. They are also less expensive to operate than conventional gasoline engines and have smooth operation controls [1,2]. Furthermore, it is estimated that over 35 million EVs will be in use worldwide by 2022 [3]. Nevertheless, a high infiltration by EVs is a significant issue that affects the electricity-distribution system, leading to problems such as power-quality degradation;

increased line damming; distribution transformer failure; increased distortion; and a higher fault current [3–5], which is one of the efficient approaches to integrated local power generation such as the renewable energy source (RESs) of EVs' charging infrastructure [3,6–8]. However, the critical issue that the limitation of EVs is their battery capacity the driver can be awareness [9]. Therefore, the locations of EV charging stations should be conveniently placed and should supply fast charging (between 20 and 30 min) in order to improve the quality of service and good experience of using EVs [10]. EV drivers can now choose the most convenient and appropriate station to use based on their preferences, thanks to the advancement of charging-station networking [9,10]. The qualities of a charging station, such as its position and size, would influence an EV driver's charging behavior with respect to making the best decisions, which would also have an impact on the charging station's performance (such as the length of the queues) [9]. It is crucial to investigate how the performance of the charging station and the charging behavior of EV drivers are related to one another. Moreover, intelligent recommendations for EV charging stations in a tourism city destination should be introduced to support smart tourism [9,11].

The concepts of smart mobility and smart transportation are currently linked to the concept of sustainable tourism as they affect issues that are related to the local economies as well as the environment [12]. Moreover, the traditional transport system in tourism produces the most externalities, with a negative impact on air and noise pollution as well as traffic congestion [12,13]. Overcrowding on streets, pavement, and public transport, as well as heavy traffic, are also recognized as important negative factors for the externality of tourism [12,14]. To reduce the negative impacts and serve tourists more efficiently, innovative, mobility-related business models and services that encompass electric vehicles can contribute significantly [12]. Moreover, the seasonality of the transport demand in touristic areas results in traffic-overcrowding phenomena and extensive air pollution. At the same time, tourist areas must serve the particular needs of tourists in order to be competitive and increase their markets. EV cars and services can contribute to tourism and tourist destinations by reducing the negative impacts of demand and serving tourists efficiently.

A major problem with using EVs is that drivers still struggle to charge their vehicles due to the relative lack of filling stations and lengthy wait times. Despite the expansion of the number of EV charging stations, the publicly accessible charging network is insufficient to meet the rapidly expanding, on-demand charging requirement. Undoubtedly, such a poor charging experience increases undesirable charging costs and inefficiency, and could even worsen range anxiety among EV drivers, which hinders the spread of EVs. Therefore, it is appealing to offer intelligent charging recommendations to enhance the experience of charging for EV drivers from a variety of perspectives, such as minimizing the wait time for charging (*CWT*), decreasing the cost of charging (*CP*) and optimizing the failure rate of charging (*CFR*) to enhance the effectiveness of the global charging network.

The trend of using EVs is increasing significantly worldwide due to their numerous benefits, such as cost savings, preserving the environment, improving traffic in smart cities, and increased user satisfaction, etc. EVs are becoming popular, and the number of users in Thailand is increasing.

The Thai government's EV promotion highlights the importance of EV charging stations in many countries. Thailand is a country that has just introduced EV cars to the market, and EV charging stations can be found throughout the capital Bangkok, in the major provinces of Chiang Mai, Phuket, and Nakhon Ratchasima, as well as in tourist destinations such as Pattaya and Hua Hin [15]. Additionally, many operators have developed their own smartphone applications that allow EV users to locate, reserve, and navigate to nearby charging stations. These applications include iEA Anywhere, MEA EV, Pumpcharge, and EVolt. However, there is not a single platform that allows users to use the charging ports at any station at once. Customers of electric vehicles (EVs) claimed that "stations with DC fast chargers may be near the offices of PEA, EGAT, or MEA, and their placements demand a diversion from the route plan [16]."

Despite the fact that many governments are expanding the publicly accessible charging network to suit the constantly increasing need for on-demand charging, EV drivers are still having difficulty charging their vehicles due to overcrowded stations and high wait times [17]. Unquestionably, a poor charging experience increases unfavorable charging costs and inefficiencies, and may even exacerbate EV drivers' range anxiety, preventing the future adoption of EVs. To improve the efficiency of the global charging network, it is tempting to offer intelligent charging recommendations to enhance the EV driver's charging experience in a number of ways, including lowering the charging price (*CP*), lowering the charging wait time (*CWT*), and maximizing the charging failure rate (*CFR*). The charging recommendation problem differs from the standard recommendation tasks in two aspects [18,19]. First, the number of charging stations in each geographic location may be restricted, resulting in a possible resource rivalry among EVs. Second, depending on the battery capacity and charging power, the battery recharging process may prevent the charging place from being used for several hours. As a result, the current proposal may have an impact on future EV charging recommendations, as well as the worldwide charging network in the long run.

Although previous studies have provided insight into the impact of EVs in Thailand, a disruptive technology transition would require new research studies to provide suggestions and recommendations in accordance with a rapid transition. In the case of charging stations, EVs support sustainability in smart tourism cities.

Thailand aims to become the main electric vehicle hub in Southeast Asia by 2030. The current government operate and develop strategy during the nascent stage of EV adoption [20]. The first major challenges for EVs in supporting tourism include high investment costs and the high price of selling electricity, especially considering that the number of EV users in Thailand is still very low [21]. Second, 70% of charging stations are in central areas such as Bangkok, Nonthaburi, and Samut Prakan [18]. Moreover, charging-station usage is divided into three types: nighttime users, daytime users, and on-the-go users. The EV infrastructure in Thailand is not sufficient for purely electric vehicles.

To bridge the gap, the aim of this paper is to study the Spatio-Temporal Multi-Agent Reinforcement Learning (STMARL) (Master) framework for an intelligent EV-charging recommendation system for a smart-tourism-city case study in Chiang Mai, Thailand. The researchers evaluated the EV-charging recommendation system based on the traditional, random-selected recommendation system and a proposed recommendation system based on the STMARL (Master) algorithm. This was carried out by building a multi-agent actor-critic architecture that uses decentralized execution and centralized training to test on a simulation using real-world EV charging station data in Chiang Mai, a smart tourism city in Thailand. The results showed that the model outperformed the competition when a comparison between four indicators, *MCWT*, *MCP*, *TSF*, and *CFR*, was conducted. The core contribution of this paper is to apply the Multi-Agent Spatio-Temporal Reinforcement Learning (Master) framework for intelligent charging recommendations. The recommended EVs were used to resolve charging problems as a MARL task; MARL was used to provide recommendations for multi-objective intelligent charging stations. This paper is composed of an introduction, a literature review, an outline of the methodology, results, discussion, and a final conclusion.

## 2. Literature Review

### 2.1. Electric Vehicles (EVs)

Currently, electric vehicles (EVs) are one of the most energy-efficient vehicle technologies available and have the greatest potential for lowering energy usage. The history of EVs is extensive. However, there has been a renewed interest in EVs over the last two decades owing to environmental issues such as pollution and global warming, as well as economic problems such as a reliance on foreign fossil fuels. An EV emits fewer greenhouse gases (GHGs) across its entire life cycle, is quieter, and has no tailpipe emissions. All of these problems are driving the development of EVs [11,22,23]. There is also a variety of EVs

currently available. The pure EV, also known as a plug-in electric vehicle (PEV), has an electric motor that is driven by electricity from a grid-connected battery. The hybrid electric vehicle (HEV) is the second type, which combines an internal combustion engine (ICE) and an electric motor to charge the battery. The HEV is not designed to use the power grid to recharge its battery. The plug-in hybrid electric vehicle (PHEV), which can be charged externally from the power grid and combines an electric motor with energy from a battery, is the third type. In addition, there is frequently a second engine, such as an ICE, that runs on a different fuel. HEV and PHEV technologies allow for smaller battery packs while preserving the same range [24].

### 2.2. Trend of Using EVs

To help slow down global warming, numerous governments and international organizations have promoted the use of EVs. The International Energy Agency (IEA) and the Clean Energy Ministry (CEM) created the "Electric Vehicles Initiative (EVI)," a policy platform with thirteen country partners, in 2010 to hasten the electrification of the transportation sector [25]. Since 2010, the adoption of electric vehicles has grown exponentially. The number of electric vehicles worldwide increased from 17,000 to 7.2 million in 2019 [26]. Out of all electric vehicles, 34 million (47%) come from China. Approximately 1.7 million (or 25%) EVs are produced in Europe, compared to 1.5 million (20%) in the US [26]. As a result, EVs are growing in acceptance everywhere. Although there are direct incentives for EV users (such as tax breaks, tax credits, and subsidies for EV purchases), the significance of EV-charging infrastructure cannot be overstated [27–29]. According to Lim et al. [29] and Melliger [30], EV users may experience "range stress," or anxiety, as a result of their EV battery's low capacity and the lack of charging stations along their travel routes. In order to alleviate EV range anxiety, governments have developed regulations and financial incentives to stimulateinvestment in charging stations. According to China's goal for its EV infrastructure, there should be 12,000 centrally located charging stations and 480,000 distributed charging locations by 2020 [31]. With the aim of having seven million charging stations by 2030, France established an energy transition law for green growth in 2015 [32].

### 2.3. Elctric Vehicle Charging Station Recommendation Systems

Due to their low carbon emissions and energy efficiency, EVs have become a popular alternative in the modern transportation system in recent years. Some efforts have been undertaken to recommend charging stations for EVs [33–41]. Most of the research [33,34,36,39,40] has concentrated on advising EV drivers on where to find charging stations in order to save time. Guo et al. [40], for example, proposed using a game-theory technique to provide charging station recommendations in order to save travel and queuing time [42–45]. Additionally, Wang et al. [46] developed a fairness-aware recommendation system to cut down on idle time based on fairness requirements. In order to facilitate site recommendations, Cao et al. [39] integrated the charging reservation data into a vehicle-to-vehicle system. A different area of study [41,43–46] looked at how to manage more challenging situations, particularly when commercial benefits were taken into account. This study was not limited to the charging site recommendation issue. In order to meet shifting consumer demand, Yuan et al. [43] proposed a charging method that allowed an electric taxi to be partially charged. To provide charging and relocation suggestions for electric taxi drivers, Wang et al. [43] developed a multi-agent, mean-field hierarchical reinforcement learning framework by treating each electric taxi as a separate agent. Through the use of deep reinforcement learning—which had already become extensively used to address problems involving sequential decision-making—this maximized the cumulative rewards of the number of served orders. However, defining each EV driver as an agent was not appropriate for our task because the majority of charging requests in our work were ad-hoc and from nonrepetitive drivers.

### 2.4. Development of Electric Vehicles in Smart Tourism Cities

In recent years, electric vehicles (EVs) have become a popular alternative in the modern transportation system due to their low carbon emissions and clean energy efficiency. Several tourism destinations around the world have started to implement smart city projects with the aim of improving the standard of living of their citizens, as well as their sustainability. In order to support smart tourism sustainability, EVs have become an increasingly popular alternative for public transport and shared mobility, etc. [44]. As has been presented in previous literature, there are various types of EVs. Pure EVs have an electric motor that is driven by electricity from a grid-connected battery, while a hybrid electric vehicle (HEV) combines an internal combustion engine and electric motor with a battery. PHEV vehicles may also have a second engine, such as an ICE, which runs on a different fuel [24]. However, research has shown that EV buyers may experience "range anxiety". They may be concerned about the limited capacity of their batteries and the availability of charging stations on their driving routes [45,46]. To address this problem, government policies and incentives have been developed in many countries to promote investment in charging stations for electric vehicles (EVs). In 2015, France enacted a law on energy transition for green growth, and it aims to have 7 million charging points by 2030 [47]. Meanwhile, China set a vision for its EV charging infrastructure, stating that there would be 12,000 centralized charging stations and 480,000 distributed charging points by 2020. Furthermore, the study of the development of electric vehicle (EV) charging stations in Thailand between 2015 and 2020 indicated that the high upfront investment costs, small number of EV users, and high electricity prices make the operators "wait-and-see." Moreover, charging station usage is divided into three types: nighttime users, daytime users, and on-the-go users. The EV infrastructure in Thailand is not sufficient for purely electric vehicles. The government has tried to address the constraints by setting up a national EV policy committee to accelerate EV adoption and EV charging stations by 2030 [25]. However, more efforts are needed to facilitate the charging process as well as to improve batteries. Currently, researchers are working on improved battery technologies to increase driving range and decrease charging time, weight, and cost. These factors will ultimately determine the future of electric vehicles (EVs) in the world [47].

### 2.5. Spatio-Temporal Multi-Agent Reinforcement Learning (STMARL) (Master)

In order to resolve the EV-charging recommendation problem as a Multi-Agent Reinforcement Learning (MARL) job, the Spatio-Temporal Multi-Agent Reinforcement Learning (STMARL) (Master) method was created. Multiple agents are involved in MARL, a novel subfield of reinforcement learning. Additionally, it assumes that rather than using traditional reinforcement learning, agents will learn to cooperate and compete with one another. Learning agents individually is the easiest way to implement a multi-agent system [48–50]. The independent actors, on the other hand, are unable to coordinate their actions and hence fail to establish intricate cooperation [51–53]. In addition, learning communication across numerous agents is a natural way to create agent collaboration [53–57]. However, due to the massive volume of data transferred, such systems always result in a high communication overhead. Alternatively, several works [57–59] have made use of a centralized-training, decentralized-execution architecture to achieve agent coordination and cooperation. In a large-scale agent system, the advantage of such methods is that the agents can perform decentralized execution without engaging in any other agents' knowledge, making them lightweight and fault-tolerant. Additionally, a few studies have successfully used MARL for a variety of intelligent transportation tasks in recent years. For example, the MARL algorithm was used by Wang et al. [60] and Wei et al. [61] for cooperative traffic signal regulation. Likewise, Jin et al. [62], Li et al. [63], Lin et al. [64], and Zhou et al. [65] used MARL to optimize the long-term benefits of a large-scale ride-hailing business. MARL has also been employed for the repositioning of shared bikes [66] and the prompt scheduling of delivery services [67,68]. However, we contend that, because our work is a recommendation, it

differs fundamentally from the aforementioned applications and that the aforementioned approaches cannot be directly applied to our issue.

In multi-agent reinforcement learning, the agent is the component that decides what action to take. To make this decision, the agent may use any observation from the environment as well as any internal rules that it has. Those internal rules can be anything, but in reinforcement learning it is typically expected that the environment will provide the current state and that the state will have the Markov property. It processes that state using a policy function $\pi(a \mid s)\pi(a \mid s)$ that decides what action to take. Furthermore, in reinforcement learning, we usually care about how to handle a reward signal (received from the environment) and will optimize the agent to maximize the expected reward in the future. To accomplish this, the agent will keep some data that is influenced by previous rewards and use it to build a better policy. The boundary of agent and environment is usually considered to be very close to the abstract decision-making unit, which is an intriguing aspect of the definition of an agent.

*2.6. Smart Tourism City*

Innovation and information technology are the hallmarks of a smart city, while virtual cities—which is the notion of a smart city—relate to a city with physical qualities and complex social and digital aspects by focusing on innovation and information technologies, respectively [69]. Moreover, a smart tourism city is a new component of utilizing information technology to support the tourism industry in the city [69,70]. The Chiang Mai smart city was developed in 2017 to create smart agriculture, a smart economy, smart safety, smart health, and smart tourism. In particular, the policy of smart tourism implements information and communications technology (ICT) infrastructure with the aim to support and serve the tourism industry of the country. Moreover, Chiang Mai is a smart tourism destination because tourists who travel within the destination use infrastructure, resources, and utilities that need to be shared with the local population. If the city has technology to support good living, there would be applications that could feed useful information to people in the area.

**3. Methodology**

In this section, we discuss a few crucial definitions and the EV-charging recommendation problem. We define the charge request by treating the activity of each day. When looking at a set of N charging stations $C = \{c_1, c_2, \ldots, c_N\}$, this request for a charge is described in the first definition as the *t*th request (i.e., step t) of a day with $q_t = l_t$, $T_t$, and $T_c$ t. In particular, the terms $q_t$ and $T_t$ refer to the current location, the real-world time of step *t*, and the charge request completion time, respectively. When a charging request completes the charging operation or gives up, we call it done (i.e., charging failure). This also refers to $|Q|$ as the cardinality of $Q$. The use of $q_t$ to signify the comparable EV of the $q_t$ interchangeably comprises the charging waiting time (*CWT*) that is calculated by multiplying the travel time from the charging request location $l_t$ to the target charging station *c* I by the amount of time needed to wait in line at *c* I until the charging request is finished. The charging price (*CP*) is calculated by the charge price using the unit price per kilowatt-hour (kWh). The cost of electricity and the service charge are typically combined to create the charging price. The percentage of charging requests that accepted our recommendation but did not complete the transaction as opposed to the total number of charging requests that did so is known as the charging failure rate (*CFR*). A sample solution for EV charging as follows: consider a day's worth of charging requests; our task is to match each request with the best charging station, or *rct*, in order to reduce the total *CWT* over the long term and average the *CP* and *CFR* for the $q_t$ Q who accept our recommendation. Finally, the limitation of this method (Multi-Agent Reinforcement Learning: STMARL) that it is stable when using a function approximation with a policy that must be implement.

### 3.1. Electric Vehicles (EVs) in Thailand

In Thailand, EVs have recently been proposed and are becoming popular; therefore, the government is also keen to increase the use of EVs and to build an EV manufacturing industry. As a result, the Thailand Board of Investment has offered a number of incentives to entice foreign direct investment in the EV industry, including a zero-percent import charge on EV-related machinery and tax benefits lasting up to eight years. A pilot project to construct charging stations and provide funding to support the installation of 150 charging stations for both public and private businesses has been established by the Ministry of Energy and the Electric Vehicle Association of Thailand (EVAT) in terms of EV charging infrastructure. Thailand's number of charging stations has continuously expanded since 2015, reaching 2285 in September 2021 [33]. Moreover, the Thai government has attempted to solve the EV-related issues by creating the National Electric Vehicle Policy Committee to encourage the use of EVs and the construction of charging stations. The Committee also established a reduced and fixed price for power for charging stations [20,30].

Chiang Mai is Thailand's second largest province. It is situated in the country's Northern area and is a business and cultural center with a population density of 1533 people/km$^2$ [30]. Additionally, Chiang Mai is a fascinating travel destination. Tourists inflate the population of the main city area, thus necessitating an efficient mass-transportation infrastructure. Regarding the government's EV promotion initiative, Bangkok and other important regions including Chiang Mai, Phuket, and Nakhon Ratchasima all have EV charging stations [20]. Chiang Mai has also been selected as the research location for a pilot study for other projects. GridWhiz assisted PEA-Encom with its EV charging operation and took part in the Chiang Mai electric tuk-tuk project. This start-up was founded in 2013 and created the pump-charge EV-charging-management software, which could discover and reserve charging outlets as well as guide EV users to appropriate charging stations [33]. The second start-up was Evolt, which made an effort to install EV chargers in locations such as government and business buildings, colleges, retail centers, and residences. Whizdom 101, Thammasat University, and the Department of Industrial Promotion (in Bangkok, Chiang Mai, and Khon Kaen) are some of their charging stations [34]. Evolt also created its own charging station locator and reservation application. They give registered users a free charge to encourage them to use their charging stations.

### 3.2. Research Framework

There are two ways that the charging recommendation challenge is different from conventional recommendation jobs. First, there may be a shortage of charging stations in a particular area, creating rivalry for EV resources. Second, the battery-recharging method may restrict the charging spot from being used for several hours, depending on the battery capacity and charging power. As a result, the current idea might have an effect on recommendations for EV charging in the future and, eventually, the global charging network. The second recommendation system in this study was implemented by applying MARL, which is known as the STMARL (Master) framework, for an intelligent charging recommendation for a smart tourism city. The first recommendation system in this study was implemented by random selection, which was a traditional method. Furthermore, the current development of deep reinforcement learning has great potential for the long-term improvement of the charging experience in a number of ways. Four parameters—the mean charging wait time (*MCWT*), mean charging price (*MCP*), total saving fee (*TSF*), and charging failure rate (*CFR*)—that could improve the effectiveness of the global charging network were examined between two random selection algorithms and a master algorithm. The research framework is displayed in Figure 1.

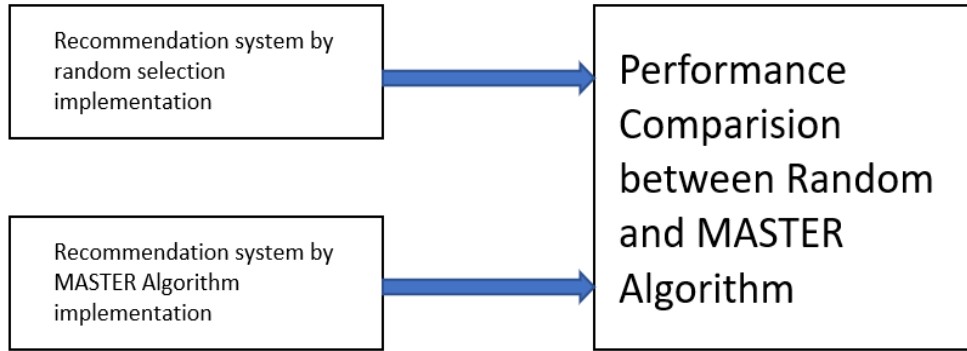

**Figure 1.** Research framework.

The EV-charging recommendation problem differs from conventional recommendations. First, there may be a shortage of charging stations in the target location, which could lead to EVs competing for limited spots and other resources. Second, the battery recharging process may obstruct the charging place for several hours depending on the battery capacity and charging power. As a result, the current recommendation might affect suggestions for EV charging in the future and have a long-term impact on the world's charging infrastructure.

In the past, certain efforts were made to recommend the best charging station for the current EV driver for each step of a single objective, such as minimizing the overall *CWT*. From a global viewpoint, however, such an approach ignores the long-term conflict between the space-constrained charging capacity and the spatio-temporally unbalanced charging demands, which results in less-than-ideal recommendations (e.g., longer overall *CWT*s and higher *CFR*s). Recent studies on order dispatching for ride-hailing and shared bike rebalancing demonstrate the enormous benefits of reinforcement learning (RL) in improving sequential decision problems in a dynamic environment. The agent in RL learns the strategy to obtain the overall, ideal, long-term reward by interacting with the environment. Consequently, it can enhance RL-based charging suggestions with long-term objectives like lowering the total *CWT*, the average *CP*, and the *CFR*.

For the EV charging recommendation task, the researchers described the concept of the MASTER Algorithm. The agent involved each charging station as an individual agent. Each agent would make timely recommendation decisions for a sequence of charging requests that would be constant throughout the day with multiple long-term optimization goals.

The observation of the agent was a combination of the index of the real-world time, the number of current available charging spots, the number of imminent charging requests (future demand), the charging power, the estimated time of arrival (*ETA*) at the location, and the *CP* at the next ET. Each agent would make timely recommendation decisions for a sequence of charging requests that would be constant throughout the day with multiple long-term optimization goals. In our MASTER Algorithm, we provided a reward-settlement mechanism (i.e., incentives were returned after completing a billing request). The multi-agent actor–critic system was developed with a centralized attentive critic for learning deterministic rules to encourage the agents to offer recommendations together. The goal of the work on recommended electric vehicle charging was to concurrently lower the total *CWT*, average *CP*, and *CFR*.

## 4. Results

The researcher put the Master to the test on the data sets in Chiang Mai. The information on the charge requests was gathered using the Google Maps API. Grids of 1 km × 1 km were used to measure the total number of potential 10-min charging requests in the vicinity. The station was in Chiang Mai, and there were eight nearby grids in addition to an anticipated future demand for charging stations. The first ten days of the data were used for training, the following three days for validation, and the final five days for testing. All of the real-world data were put into an EV-charging recommendation simulator for the experiment. All tests were performed on an Intel(R) I5 processor clocked at 2.50 GHz. We chose d = 30 min, temperature = 0.2 for the updated weight modification, and discount factor = 0.99 for learning all the RL algorithms while charging the competition modeling. Three linear network layers with 64 and hidden levels that made use of the ReLU activation function were present in both the actor and critic networks. The target network's soft update was set at 0.001, the replay buffer size was 1000, and the batch size was 32. We utilized the Adam optimizer for all programmable algorithms to train our model with the learning rate set to 50,000. We adjusted each baseline's major hyperparameters using a grid search methodology. With the best iteration chosen by the validation set for testing after fifty iterations, the RL algorithms were trained to recommend the top ten nearby charging stations.

Moreover, the researchers developed four indicators to assess how well our strategy and recommendation systems performed. $Q^a$ was defined as the number of charge requests that agreed to our suggestions. We defined $Q^s \subseteq Q^a$ as the set of charging requests that accepted our advice and were charged successfully. The cardinalities of $Q^a$ and $Q^s$ were $|Q^a|$ and $|Q^s|$, respectively. In order to assess the overall charging wait time of our recommendations, we also defined the mean charging wait time (*MCWT*) over all charging requests $q^t \in Q^a$ (Equation (1)) [71].

$$MCWT = \frac{\sum qt \in Q^a CWT(q_t)}{|Q^a|} \tag{1}$$

The researchers calculated the mean charge price (*MCP*) over all charging requests, where *CWT* $(q^t)$ represented the charging request's wait time (in minutes). We determined the mean charging price (*MCP*) over all the charging requests in order to assess the average charging price $q^t \in Q^s$ (Equation (2)):

$$MCP = \frac{\sum qt \in Q^s CP(q_t)}{|Q^s|} \tag{2}$$

where *CP* $(q^t)$ was the charge price of *qt* (in *CNY*).

By comparing our recommendation algorithm with the ground-truth billing activities, we developed the total saving fee (*TSF*) to evaluate the average of total fees saved every day.

*RCP* $(q_t)$ represented the real-world charging action charging rate, *CQ* $(q^t)$ represented the *qt* electric charging amount, and *Nd* represented the number of evaluation days. Additionally, it was important to keep in mind that the *TSF* could have a negative value, which would show how many fees were overspent in comparison to the actual billing actions. Finally, we established the charging failure rate to assess the ratio of the charging failures in our recommendations (CFR) (Equation (3)):

$$TSF = \frac{\sum qt \in Q^s (RCP(q_t) - CP(q_t)) \times CQ(q_t)}{Nd} \qquad (3)$$

where $CQ(q_t)$ is the $qt$ electric charging quantity, $Nd$ is the number of assessment days, and $RCP(q_t)$ is the ground-truth charging-action charging price. It was important to remember that the $TSF$ may have a negative value, which would show how many fees were paid in excess of what was actually charged. Finally, we created the $CFR$ (Equation (4)) to assess the percentage of charging failures in our recommendations:

$$CFR = \frac{|Q^a| - |Q^s|}{|Q^a|} \qquad (4)$$

*4.1. Overall Performance*

Table 1 shows the overall outcomes of our methodology as well as all the baselines that were compared in two data sets in terms of our four metrics. Overall, the Master delivered the most well-rounded performance out of all the baselines. When compared to the ground-truth charge activities in Chiang Mai, the Master was reduced (58.9%, 9.3%, and 95.4% for the *MCWT*, *MCP*, and *CFR*, respectively).

**Table 1.** Overall performance evaluated by *MCWT*, *MCP*, *TSF*, and *CFR* in Chiang Mai.

| Algorithm | MCWT | MCP | TSF | CFR |
|---|---|---|---|---|
| Random | 37.64 | 1.728 | −349 | 48.3% |
| MASTER | 15.46 | 1.567 | 15761 | 2.2% |

The impact of the top-k active agent concentration on the Cheang Mai suggestions was investigated by the researchers. We increased k from 25 to the overall number of agents. As can be observed, lowering the recommendation restriction k increased the *MCWT* while increasing the *MCP* and *TSF*, illustrating how the best solutions for the various goals diverge. This made sense because a larger pool of applicants would indicate farther-flung, more affordable charging stations. The *MCWT* and *MCP* were 9.98; 1.649 and 11.72; 1.491, respectively, which were not severe and still adequate for online recommendation. However, the performance with the toughest constraint (i.e., k = 20) and without constraint differed only marginally. In terms of the candidate numbers, the previous statistics suggested that our model was well-balanced (Figure 2). This also motivated us to consider how, in the future, we could be able to deliver a variety of recommendations that would be slanted toward different aims in order to meet the specific preferences.

Each actor is a charging station $(c^i)$ which is an individual agent. Each agent will decide on recommendations for a series of charging requests. The observation $o_t^i$ of agent $(c^i)$ is a combination of the index of $(c^i)$, the number of current available charging station $(c^i)$ and the number of charging requests around $(c^i)$ for future charging demand. Each agent $(c^i)$ offers a bid value for $q_t$ as its action $(a_t^i)$ that is the joint action, and $q_t$ will be used to suggest for the agent with the highest bid value. The parameter $p_t^i$ represents upcoming information about the billing and competitive future of each $(c^i)$ for $q_t$, so the parameter $p_t$ is the future knowledge of active agents that each agent's policy $(c^i)$ can be updated by the gradient $\nabla_{\theta_b^i} J(b^t)$.

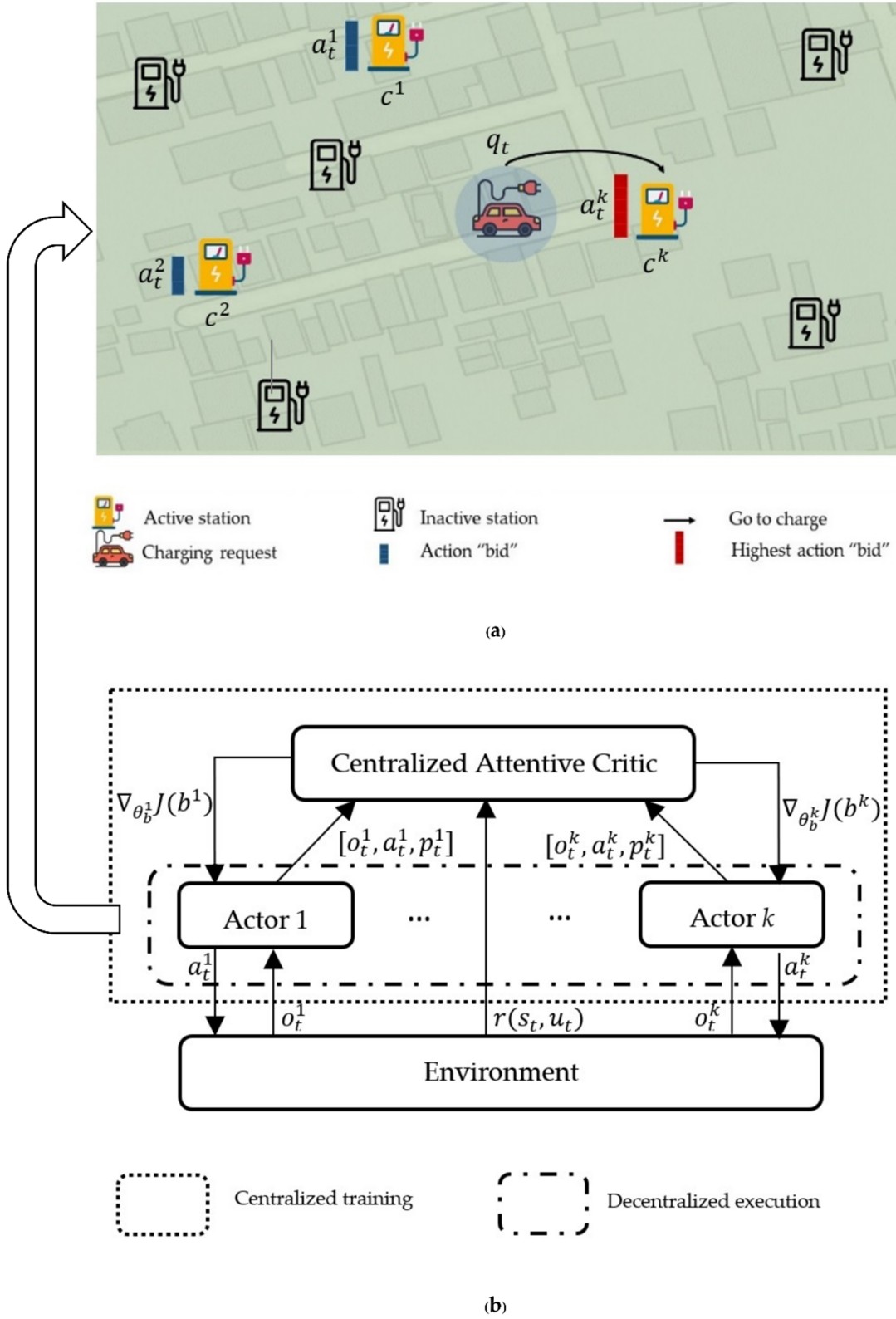

**Figure 2.** The implementation of electric vehicle (EV) charging recommendation systems. (**a**) EV Charing Station (**b**) EV Charing Framework.

## 4.2. Implication of EVCs

When the EVC system was implemented in Chiang Mai, the charging stations with high action values were visible to the researchers, who paid close attention to them (Fig-

ure 3). This was evident, given that there were many bidders for these charging stations. The most active charging station would receive the charging request, and the ecosystem would reward this recommended station in kind. Additionally, we observed that the action was strongly correlated with the supply, future supply, demand, *ETA*, and *CP*. A charging station with a high action value had a low *ETA* and *CP* and had enough available charging places (supply). Charging stations with a low number of available charging spots but a high future demand, on the other hand, typically had a poor action value for avoiding future charging competition. Therefore, the above findings supported our model's ability to manage the conflict between space-constrained charging capacity and spatially and temporally unbalanced charging requests.

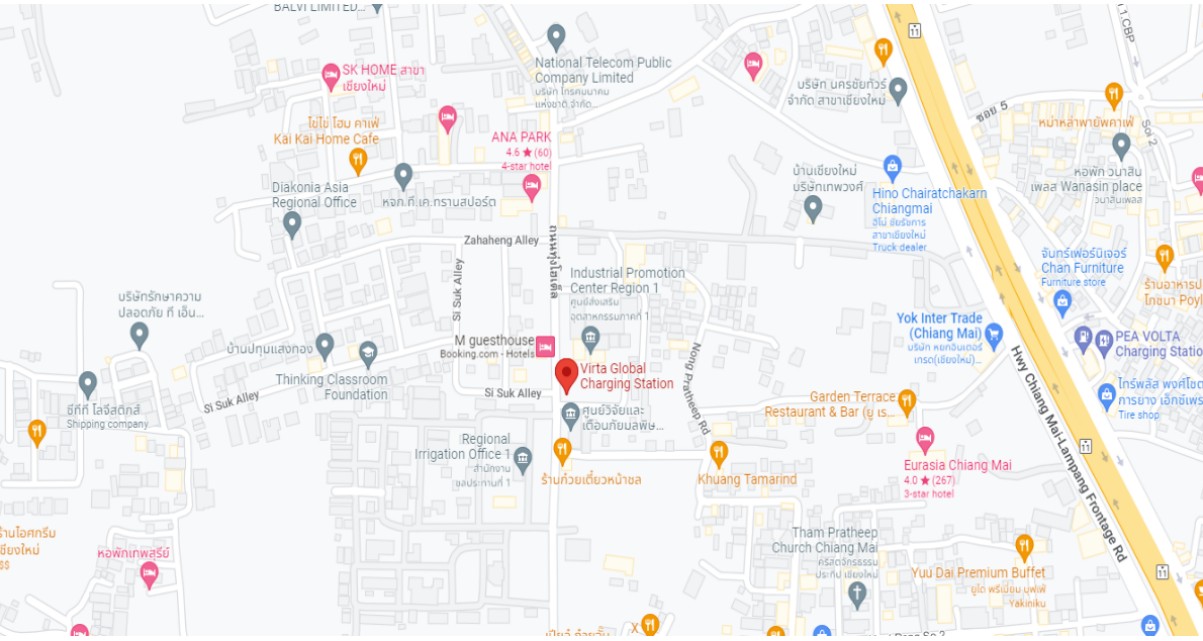

**Figure 3.** The implementation of electric-vehicle-charging recommendation systems in Chiang Mai.

*4.3. User's Evalutuion of EVCs*

Following the implementation of the EVC system in Chiang Mai, the system satisfaction has been evaluated by users, both Thai and international tourists. The data collection results on the satisfaction of 200 tourists using the smart charging stations service were divided into 100 Thai and 100 foreign tourists. Table 2 shows that Thai tourists were satisfied at the highest level ($\overline{X}$ = 4.26, S.D. = 0.70). The number one demand for the EVC system had the highest average level of agreement ($\overline{X}$ = 4.34, S.D. = 0.64). On the second, the utilization of EVCs had the highest average level of agreement ($\overline{X}$ = 4.32, S.D. = 0.74), and the third place evaluated user confidence in EVCs to promote a sustainable smart city. The mean had the most agreeable level ($\overline{X}$ = 4.21, S.D. = 0.75), and the fourth regarded developing the EVC system to promote tourism. The mean values were at the high level of agreement level ($\overline{X}$ = 4.18, S.D. = 0.65), respectively.

Moreover, the overall foreign tourists were satisfied at a high level ($\overline{X}$ = 4.18, S.D. = 0.85). The number one demand for the EVCs system had an average level of agreement at the highest level ($\overline{X}$= 4.28, S.D. = 0.91). The second rank—the utilization of EVCs—had the highest average level of agreement ($\overline{X}$= 4.21, S.D. = 0.88), and the third rank regarded the development of the EVC system to promote tourism. The average level was very agreed ($\overline{X}$ = 4.18, S.D. = 0.75). Finally, the fourth place reported confidence in EVCs to promote smart cities towards sustainability. The average scores were at the very high agreement level ($\overline{X}$ = 4.04, S.D. = 0.86), respectively.

**Table 2.** User evaluation of EVCs.

| Item | Thai Tourist (*n* = 100) | | International Tourist (*n* = 100) | |
|---|---|---|---|---|
| | $\overline{X}$ | S.D. | $\overline{X}$ | S.D. |
| The demand for the EVC system | 34.4 | 0.64 | 4.28 | 0.91 |
| The utilization aspect of EVCs | 4.32 | 0.74 | 4.21 | 0.88 |
| EVC system development to promote tourism | 4.18 | 0.65 | 4.18 | 0.75 |
| Confidence in EVC to promote sustainable tourism | 4.21 | 0.75 | 4.04 | 0.86 |
| Total average | 4.26 | 0.70 | 4.18 | 0.85 |

Table 3 demonstrates the ANOVA test comparing the services of intelligent charging stations between Thai and foreign tourists. By testing with a *t*-test, it was found that Thai and foreign tourists exhibited no difference in satisfaction levels with using EV smart charging stations.

**Table 3.** ANOVA test.

| User | *n* | $\overline{X}$ | S.D. | *t* | *p* |
|---|---|---|---|---|---|
| Thai tourist | 100 | 4.26 | 0.70 | −0.068 | 0.946 |
| International tourist | 100 | 4.18 | 0.85 | | |

$p < 0.05$.

The analysis of the results concerning the comparison of international and domestic tourist satisfaction found that there was no significant difference in using EV smart charging stations in Chiang Mai in terms of the demand for using EVC stations, the EV car utilization aspects, EVC system promotion in tourism, and the confidence in using EVCs to promote sustainable tourism. This can create a new business model to provide mobility services as well as more EVC stations to serve tourists at tourist destinations, creating positive tourism experiences and value that can lead to revisiting.

Moreover, as a result of tourists using EVC recommendation systems in Chiang Mai, electric vehicle charging stations have become convenient for tourists, provide easy access to tourist attractions, and provide a means to travel safely and efficiently. They also increase the efficiency of urban planning in the smart city to support smart tourism, which will create satisfaction, positive impressions and the desire to return to travel again. EVCs will generate income and to conserve the environment, which eventually leads to sustainable tourism and enhances the sustainable development of the SDG goals.

## 5. Conclusions & Discussion

This paper aims to improve the efficiency of the global charging network. It would be beneficial to offer intelligent charging recommendations to enhance the charging experience of EV drivers in a number of ways, including lowering the charging price (*CP*), lowering the charging wait time (*CWT*), and maximizing the charging failure rate (*CFR*).

The EV is a disruptive technology transition that has been become widespread worldwide in order to reduce pollution and traffic congestion while increasing cost savings and preserving the good image of a tourist destination [12–14]. The research gap is the lack of innovation and prototypes for the EVC system to support smart tourism. To bridge the gap, this paper aims to study the Spatio-Temporal Multi-Agent Reinforcement Learning (STMARL) (Master) framework for an intelligent, EV-charging recommendation system using a smart tourism city case study in Chiang Mai, Thailand.

This study added to the understanding of the STMARL (Master) framework for spatial-temporal, multi-agent reinforcement learning for an intelligent electric vehicle (EV) charging recommendation system for the smart tourism city case study of Chiang Mai,

Thailand. The researchers examined the intelligent EV-charging recommendation system with the long-term objective of lowering the total *CWT*, average *CP*, and *CFR*. In this study, several parameters were optimized. A STMARL (Master) framework was suggested, and this subject was defined as a multi-objective MARL challenge. By considering each charging station as an autonomous agent, the creator of the multi-agent actor–critic framework with a centrally attentive critic hoped to encourage the agents to learn coordinated and cooperative policies.

Each agent would act as an EV charging station and attempt to find an optimal solution for the maximum reward to help the recommendation solve the problem effectively when compared to random selection. The results show an intelligent EV-charging station recommendation for tourist cities.

To increase the efficacy of the recommendation, the researchers also created a delayed-access technique to incorporate knowledge about future charging competitions during the model training. This was possible by integrating the model with a dynamic gradient-reweighting technique to adaptively influence the optimization direction of numerous diverging recommendation targets, which also expanded the centralized attentive critic to include multiple critics. The recommendation system could help individual electrical vehicles find an optimal charging station in terms of distance and charging cost.

Additionally, based on the locations of the current EV charging stations in Chiang Mai, the researchers developed the STMARL (Master) framework for intelligently suggesting publicly accessible charging stations to reduce the waiting time and charging cost. This would be undertaken effectively compared to the random selection of an EV charging recommendation that could be applied to other smart tourism cities. The proposed recommendation system was capable of electricity pricing management in order to save the cost of charging, which was an objective in the STMARL framework.

The EV charging recommendation system could thus support the increasing number of EVs and charging stations in other smart tourism cities in the future [69,70], especially after the post COVID-19 period, for which tourism would have developed resilience. Consequently, EVs would be another choice for rental and travel by a free, individual traveler (FIT) [70–76].

Moreover, the EVC system has been implemented and evaluated by both international and domestic tourists. The results found that there was no significant difference in using EV smart charging stations in Chiang Mai in terms of the demand for using EV stations, the EV car-utilization aspects, the EVC system promotion in tourism, and the confidence of using EVCs to promote sustainable tourism [12–14]. Moreover, it leads to the creation of a new business model [13] and the generation of income as well as the conservation of the environment, which leads to sustainable tourism and enhancement of sustainable development [77,78].

The limitation of this study is that the sample size is not large enough because the amount of EVs in Thailand is much less than in other countries, such as China. Finally, further research of this work could be extended to include energy-network management to sustain energy for Thailand and with a comparison with other optimization algorithms to improve our system [77,78].

**Author Contributions:** Conceptualization, P.S. and P.J.; methodology, P.S. and P.J; software, P.S. and P.J.; validation, P.S. and P.J.; formal analysis P.S. and P.J.; investigation, P.S. and P.J, resources, P.S. and P.J; data curation, P.S. and P.J.; writing—original draft preparation, P.S. and P.J.; writing—review and editing, P.S., P.J., P.K. and C.N.; visualization, P.S. and P.J.; writing—review and editing, P.S., P.J., P.K. and C.N.; supervision, K.J.; project administration, P.S.; funding acquisition, P.S. All authors have read and agreed to the published version of the manuscript.

**Funding:** This research was funded by Suan Dusit University under the Ministry of Higher Education, Science, Research and Innovation, Thailand, grant number 65-FF-003 Innovation of Smart Tourism to Promote Tourism in Suphan Buri Province.

**Institutional Review Board Statement:** The study was conducted in accordance with the ethical and approved by the Ethics Committee of Suan Dusit University (SDU-RDI-SHS 2022-030, 1 June 2022) for studies involving humans.

**Informed Consent Statement:** Not applicable.

**Data Availability Statement:** Not applicable.

**Acknowledgments:** The research team would like to thank Suan Dusit University for the funding support and would also like to thank the Chiang Mai Municipality for all their cooperation and providing the necessary information for the research.

**Conflicts of Interest:** The authors declare no conflict of interest.

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
