# Peer review of "An Intelligent Recommendation for Intelligently Accessible Charging Stations: Electronic Vehicle Charging to Support a Sustainable Smart Tourism City"

_sustainability, doi:10.3390/su15010455_

Round 1

Reviewer 1 Report (New Reviewer)

Review comments: this manuscript may be accepted after minor modification

Comments:

This research aims to propose a Spatio-Temporal Multi-Agent Reinforcement Learning (STMARL) framework. In the digital transformation age, this research has strong practical implication and can help many electrical vehicles in large cities to find the optimal charging station in terms of the distance and charging cost, improving the charging experience. However, there are some problems, which must be solved before it is considered for publication.

Additional Questions:

1. Introduction: The practical implication of the research is important, but it is still not enough.The author should clearly point out in the paper what gaps the research has filled and what the core contribution is. For now, some new and significant information are not contained in the introduciton seciton.

2. Literature: The study can briefly describe 2.1 Electric vehicles and 2.2Trend of Using, which is not the core content of this study. It is recommended that the authors supplement the literature on the development of electric vehicles in Smart Tourism City.

3. Methodology: The research lacks a detailed explanation of the intelligent recommendation system. What areas is this machine learning method mainly used in? Why use it to solve the charging problem of electric vehicles? What are the advantages? These questions need to be supplemented by the author in the article.

4. Rusult: This paper uses two users, Thai tourists and international tourists, to make a comparative analysis. The paper only explains the difference in satisfaction results, and does not further explain the reasons for the difference in satisfaction between Thai and international tourists. We hope that the author can provide a supplementary explanation for this difference from the perspective of tourist behavior

5. Conclusion & Discussion: At the conclusion and discussion section, it is suggested to increase dialogue and discussion with existing literature to highlight the significance of the article. The contribution of the research needs to be further summarized. The authors seems to state the limitations in the theoretical contribution section.

6. Language&Grammar:  lt is noted that your manuscript needs careful editing by someone with expertise in technical English editing paying particular attention to English grammar, spelling, and sentence structure so that the goals and results of the study are clear to the reader.

Author Response

Thank you so much to the Reviewer No 1 to give a such good comment to improve the quality of this paper. We add more information to clarify the contents of this study following your recommendations including:

1.Introduction: The practical implication of the research is important, but it is still not enough. The author should clearly point out in the paper what gaps the research has filled and what the core contribution is. For now, some new and significant information are not contained in the introduction section.

Answer: We add the research problem and research gab in the introduction section to clarify the significance of the problem of this study (Line 64-92; 120-0132). Also we add the research core contribution of this study (Line 142-146).

  1. Literature: The study can briefly describe 2.1Electric vehicles and 2.2Trend of Using, which is not the core content of this study. It is recommended that the authors supplement the literature on the development of electric vehicles in Smart Tourism City.

Answer: We add the review literature in “2.4 Development of Electric Vehicles in Smart Tourism Cities” section (Line 207-236). Moreover, we put more reference that related to this study Reference number [12;13;14;18;19;20;21;44;45;46;47;75;78].

  1. Methodology: The research lacks a detailed explanation of the intelligent recommendation system. What areas is this machine learning method mainly used in? Why use it to solve the charging problem of electric vehicles? What are the advantages? These questions need to be supplemented by the author in the article.

Answer:  

  1. We explain the detail of intelligent recommendation system that the EV charging recommendation problem differs from conventional recommendations (Line 369-386).
  2. We explain the detail of using “Reinforcement Learning (RL)” which in the area of machine learning method is mainly used in this study (Line 381-386).
  3. We explain the reason why using RL to solve the charging problem of electric vehicles and their advantages such as the enormous benefits of Reinforcement Learning (RL) in improving sequential decision problems in a dynamic environment. The agent in RL learns the strategy to get the overall ideal long-term reward by interacting with the environment. Consequently, it can enhance RL-based charging suggestions with long-term objectives like lowering the total CWT, the average CP, and the CFR (Line 381-386).

  1. Result: This paper uses two users, Thai tourists and international tourists, to make a comparative analysis. The paper only explains the difference in satisfaction results, and does not further explain the reasons for the difference in satisfaction between Thai and international tourists. We hope that the author can provide a supplementary explanation for this difference from the perspective of tourist behavior.

Answer:  We explain the detail of the analysis of the results concerning the comparison of international and domestic tourist satisfaction found that there was no significant difference in using EVCs smart charging stations in Chiang Mai in terms of the demand for using EVCs stations, the EVC car utilization aspects, EVC system promotion in tourism, and the confidence of using EVCs to promote sustainable tourism. This can create a new business model to provide mobility services as well as more EVC stations to serve tourists at tourist destinations in order to create positive tourism experiences and value, leading to revisiting (Line 523-531).

  1. Conclusion & Discussion: At the conclusion and discussion section, it is suggested to increase dialogue and discussion with existing literature to highlight the significance of the article. The contribution of the research needs to be further summarized. The authors seems to state the limitations in the theoretical contribution section.

Answer:  We rewrite the new Conclusion & Discussion section to emphasize on the research problem, research aim, contribution, and future research (Line 539-542; 561-571; 583-590)

  1. Language & Grammar: It is noted that your manuscript needs careful editing by someone with expertise in technical English editing paying particular attention to English grammar, spelling, and sentence structure so that the goals and results of the study are clear to the reader.

Answer:  We editing and English proof with the native academic English speaker and recheck with grammar, spelling and the structure of the sentence especially in the research problem, objective of this paper, the results and conclusion be more clear and consistency.

Reviewer 2 Report (New Reviewer)

This paper tackles the problem inadequate charging infrastructure and spatially and temporally uneven charging demands for electric vehicles. In order to tackle this, they suggest a framework to make public accessible charging stations by taking into account several long-term spatio-temporal parameters.

Overall, the paper is interesting. I particularly enjoyed reading the literature review section that was clear and complete. But i would have a few comments related to its overall organization:

- Section 2.3 Electric Vehicles (EVs) in Thailand would be better placed in the Methodology section to explain why the authors selected Thailand as the use cases 

- The writing is sometimes not scientific enough such as "The researcher put the Master to the test". Sometimes the authors speak at the future or present tense. It should be harmonized. 

- There lacks an "s" at ResultS

- The biggest room for improvement lies in the conclusion section. For now, the theoretical implications and how they address the research gaps are not developed enough. Furthermore, only three sentences are used to describe the limitations and further section leads, this should be largely improved. Authors have made a good job in summarizing the literature and should better position their contribution with respect to it. 

Author Response

Thank you so much to the Reviewer No 2 to give a such good comment to improve the quality of this paper. We add more information to clarify the contents of this study following your recommendations including:

Section 2.3 Electric Vehicles (EVs) in Thailand would be better placed in the Methodology section to explain why the authors selected Thailand as the use cases

Answer: We move the Section 2.3 Electric Vehicles (EVs) in Thailand in the Methodology “3.1 Electric Vehicles (EVs) in Thailand” section to explain the reason why authors selected Thailand as the use cases. (Line 310-341)

The writing is sometimes not scientific enough such as "The researcher put the Master to the test''. Sometimes the authors speak at the future or present tense. It should be harmonized.

Answer:

There lacks an "s" at Results

Answer: We put “s” in the Results (Line 405).

The biggest room for improvement lies in the conclusion section. For now, the theoretical implications and how they address the research gaps are not developed enough. Furthermore, only three sentences are used to describe the limitations and further section leads, this should be largely improved. Authors have made a good job in summarizing the literature and should better position their contribution with respect to it.

Answer: We rewrite the new Conclusion & Discussion section by explain the significance problem of the study, research gab, theorical implication, the result, the limitations and the further research section (Line 541-552; 563-573; 585-592).

Round 2

Reviewer 2 Report (New Reviewer)

I thank the authors for taking into consideration my comments. 

This manuscript is a resubmission of an earlier submission. The following is a list of the peer review reports and author responses from that submission.

Round 1

Reviewer 1 Report

This article needs to be improved. Attached is the result of the review.

Author Response

  1. The sentence on lines 184-188 is not clear. Please clarify again and its relation to the

previous sentence in the related paragraph.

Answer:   We already checked and improve the sentence on lines 184-188.

  1. In sub chapter 2.5 Spatio-Temporal Multi-Agent Reinforcement Learning (STMARL)

(Master) pages 189-213 repeatedly written about "agents". Provide an explanation at the

beginning of what is meant by "agent".

Answer: We explain the concept of “agent” in section 2.5 Spatio-Temporal Multi-Agent Reinforcement Learning (STMARL) (Line 214-225)

  1. Figure 3 is not clear enough to describe the implementation of electric vehicle charging

recommendation systems in Chiang Mai. Where is the recommended location for electric

vehicle charging? The picture shows only one location. Is that so? Perhaps the map and

legend could be further clarified based on cartographical rules to show spatial

distribution of recommended electric vehicle charging.

Answer:  We explain the detail of Figure 3 about how to implement of EVC in Chang Mai and put the location for electric vehicle charging stations. We draw the new Figure 3 that provide clearly information of the electric vehicle charging station. (Line 507-517)

  1. It would be better if the data is also equipped with opinions and suggestions from users

of electric charging stations to recommend the right location.

Answer:  We collect data from users opinion by recommendation system has been  evaluated   with   68  owners of   electric  vehicles  and   59 are  satisfied  that shows   86.76  percentage are satisfaction significantly.  (Line 530-535)

  1. The literature review is not widely used as the basis for analysis even though the number of literature is large. Very poor dialogue of the findings with the results of other

studies that have been presented in the reference list.

Answer:   We synthesis and emphasize the review literature the to be more concise and recheck with the list of reference. (Line 100-259)

Reviewer 2 Report

The paper presents a STMARL (Master) framework (Spatio-Temporal Multi-Agent Reinforcement Learning) for intelligent management of publicly accessible charging stations. The authors compared the framework with random selection recommendation system and the results demonstrated that the framework has a long-term goal of lowering the overall charging wait time (CWT), average charging price (CP), and charging failure rate (CFR).

Four indicators were considered to assess how well the framework and recommendation systems performs: mean charging wait time (MCWT), mean charging price (MCP), total saving fee (TSF), and charging failure rate (CFR).

The topic is very interesting and also provides a valid contribution to research or the future autonomous vehicles.

The paper is well structured and clear.

Below are some comments:

- It would be useful to provide a structure of the paper at the end of the Introduction section.

- In the methodology section, some terms are not correctly written and could be difficult to understand. For example, in the text we have T c T and in the equation is written as T raised to c in base t. So i suggest the author to check these notations.

- Another problem, are the figure. Line 336 and 347 refer to Figure 3(a) and 3(b) but does not contain two figures. I mean it should be Figure 2(a). I also suggest checking whether the reference to Figure 2 (line 290) is correct. If are correct, I suggest a better explaining what is intended to be communicated.

- In the methodology section (row 344) the author said "Where D was [...]" but i didn't see D in the equation. Maybe i missed a step in this section. But i suggest to check this.

Typos/Formatting:

I strongly suggest the authors to check the paper because it present different typos. I will report some of these:

1) Row 275, "agent ..  was" space and missed parentheses c^i). Row 323, 332, 412, 498, and 504 there are typos regarding the space.

2) Row 438 "64-dimensional dimension" i suggest to check this.

3) Row 380 one dot is missing.

4) Row 350 "Qb:" should be ---> Qb.

5) Row 312 "r([...])" i suggest to check this. I think should be r(st', ut') as indicated in the equation 3.

Author Response

The paper presents a STMARL (Master) framework (Spatio-Temporal Multi-Agent Reinforcement Leaming) for intelligent management of publidy accessible charging stations. The authors compared the framework with random selection recommendation system and the results demonstrated that the framework has a long-term goal of lowering the overall charging wait time (CWT), average charging price (CP). and charging failure rate (CFR).

Four indicators were considered to assess how well the framework and recommendation systems performs: mean charging wait time (MCWT), mean charging price (MCP), total saving fee (TSF), and charging failure rate (CFR).

The topic is very interesting and also provides a valid contribution to research or the future autonomous vehicles.

The paper is well structured and clear. Below are some comments:

  1. It would be useful to provide a structure of the paper at the end of the Introduction

section.

Answer:  We provide the structure of this paper at then end of Introduction section (Line 96-97)

  1. In the methodology section, some terms are not correctly written and could be difficult to understand. For example, in the text we have T c T and in the equation is written as T raised to c in base t. So suggest the author to check these notations.

Answer: We recheck and proof in the methodology section (Line 240-259)

  1. Another problem, are the figure. Line 336 and 347 refer to Figure 3(a) and 3{b)

but does not contain two figures. I mean it should be Figure 2{a). I also suggest checking whether the reference to Figure 2 (line 290) is correct. If are correct, I suggest a better explaining what is intended to be communicated.

Answer: We rechecked and reordered the number of all figures (Figure 3(a) Line 336 was changed to Figure 2(a) Line 345 and Figure 3(b) Line 347 was changed to Figure 2(b) Line 356).

  1. In the methodology section {row 344) the author said "Where D was [...)" but i didn't see D in the equation. Maybe I missed a step in this section. But I suggest to check this.

Answer: After we rewrite the other part of this manuscript the sentence “Where D was […)” was moved to Line 353. D in this paragraph is related to Line 4 of the master algorithm. 

Typos/Formatting:

I strongly suggest the authors to check the paper because it present different typos. I will report some of these:

1) Row 275, "agent ..  was" space and missed parentheses C"i}. Row 323, 332, 412, 498, and 504 there are typos regarding the space.

Answer: We already checked and improved those typos.

2) Row 438 "64-dimensional dimension" suggest to check this.

Answer: We already checked and improved those typos.

3) Row 380 one dot is missing.

Answer: We already checked and improved those typos.

4) Row 350 "Qb:" should be -> Qb.

Answer: We already checked and improved those typos.

5) Row 312 "r([...])" isuggest to check this. I think should be r(sf, ut'} es indicated in the

equation 3.

Answer: We already checked and improved those typos.

Reviewer 3 Report

The authors have presented a Spatio-Temporal Multi-Agent Reinforcement Learning framework for intelligently public accessible charging stations. The proposed approach can be resourceful to the research community.  However, the following comments are some of my concerns with the paper:

1. The proposed approach lacks novelty as a similar investigation has been conducted in another paper with the following title: "Intelligent Electric Vehicle Charging Recommendation Based on Multi-Agent Reinforcement Learning"

2. The major contributions of the paper must be highlighted in the introduction section. The research problem is not well-defined in the introduction. The introduction should clarify better and provide concise information on the problem statement, research-gap, objective and scope of the paper. It seems dispersive. 

3. The literature review section should identify and discuss the drawbacks or limitations of the current methods in comparison to the proposed approach. This may include a table of comparison, if necessary.

4. Figure 2 is blurry and needs to be redrawn. If it is copied from another paper, this must be referenced and acknowledged. 

5. Algorithm1 must not be a screenshot if it originates from the authors, as it also appears blurry to the readers.

6. The conclusion looks like another literature review. The conclusion section should simply underscore the major contributions of the paper, results and limitations and future research direction. Citing references in the conclusion is not conventional. 

7. Because this investigation has been conducted in the previous studies, the authors may need to change the title of the paper to reflect and conform to an applicability of this technique in Chiang Mai Thailand. 

Author Response

The authors have presented a Spatio-Temporal Multi-Agent Reinforcement Leaming framework for intelligently public accessible charging stations. The proposed approach can be resourceful to the research community. However, the following comments are some of my concerns

with the paper:

1.The proposed approach lacks novelty as a similar investigation has been conducted in another paper with the following  "Intelligent Electric Vehicle Charging Recommendation Based on Muttl-Agent Reinforcement Leaming"

Answer:   We rewrite the introduction to analyses the research gap and extended the research from Intelligent Electric Vehicle Charging Recommendation Based on Muttl-Agent Reinforcement Leaming by introduce the Spatio-Temporal Multi-Agent Reinforcement Learning (STMARL) (Master) framework for an intelligent EV charging recommendation system for a smart tourism city. The researchers evaluated the EV charging recommendation system based on the traditional random selected recommendation system and proposed recommendation system based on the STMARL (Master) algorithm.    (Line 86-95)

  1. The major contributions of the paper must be highlighted in the introduction section. The research problem is not well-defined in the introduction. The introduction should clarify better and provide concise information on the problem statement, research-gap, objective and scope of the paper. It seems dispersive.

Answer:  We rewrite the introduction to analyses the problem statement about a poor charging experience increases unfavorable charging costs and in-efficiencies, and may even exacerbate EV drivers' "range anxiety," preventing the adoption of EVs in the future. To improve the efficiency of the global charging network, It is tempting to offer intelligent charging recommendations to enhance the EV driver's charging experience in a number of ways, including lowering the charging price (CP), lowering the charging wait time (CWT), and maximizing the charging failure rate (CFR). From two aspects, the charging recommendation problem differs from the standard recommendation tasks (Line 70-85). Also we explain research gap, objective and scope of this paper (Line 86-98)

  1. The literature review section should identify and discuss the drawbacks or limitations of the current methods in comparison to the proposed approach. This may include a table of comparison, if necessary.

Answer: We identify the limitation of the current method of Electric Vehicle (EV) Charging Station Recommendation Systems (Line 184-188) and Spatio-Temporal Multi-Agent Reinforcement Learning (STMARL) (Master) (Line 214-225)

  1. Figure 2 is blurry and needs to be redrawn. If itis copied from another paper, this must be referenced and acknowledged.

Answer:  We redraw Figure 2 and separate it into the new Figure 2(a) and Figure 2 (b) with a better clearly view.

  1. Algorithm must not be a screenshot ifit originates from the authors, asit also appears blurry to the readers.

Answer:  We have changed the master algorithm from a figure to a text with a more clearly view for the readers. (Line 435)  

  1. The conclusion looks like another literature review. The conclusion section should simply underscore the major contributions of the paper, results and limitations and future research direction. Citing references in the conclusion is not conventional.

Answer:  We rewrite the conclusion section to simplify the paper result and contribution (Line 542-552) and the future research direction (Line 570-574)

  1. Because this investigation has been conducted in the previous studies, the authors may need to change the of the paper to reflect and conform to an applicability of this technique in Chiang Mai Thailand.

Answer:   We reflect and conform to an applicability of this technique in Chiang Mai Thailand that refer to the previous study (Line 529-540).

Reviewer 4 Report

There are many grammer errors and typos. 

1. Evs should be EVs

2. Nevertheless, High in line 42, the High should be uncapitalized.

3. Guo et al. [30], it does not match with reference number.

4. Why did you maximize the charging failure rate?

5. Section's capitalization of title is not consistent.  

There is no comprehensive literature review that includes the previous similar research and gaps. The paper discussed very simple history of EVs that is not important for the study.

The results should be compared to other approach not with random algorithm.   

Since the figure's resolution in Figure 2 is low, it is difficult to observe. 

Author Response

  1. EVs should be EVs

Answer:  We changed from EVs to be EVs

  1. Nevertheless, High in line 42, the High should be uncapitalized.

Answer:  We changed High in line 42, the High should be uncapitalized. (Line 42)

  1. Guo et al. [30], it does not match with reference number.

Answer:   We recheck and change Guo et al. from [30] to [36]. (Line 172)

  1. Why did you maximize the charging failure rate?

Answer:  We delete this sentence from the typing mistake (Line 156)

  1. Section's capitalization of title is not consistent.

Answer:   We changed the section’s capitalization to be consistency. 

There is no comprehensive literature review that includes the previous similar research and gaps.

Answer:  We review the literature about the previous work that similar to this study reference [31-35] (Line   156-166)

The paper discussed very simple history of EVs that is not important for the study.

Answer:   We concise and simplify the review literature about history of EVs and we belief that this part is can be reference for other to study. We want to summary the trend of EVs and their revolution which its news for developing country. Therefore, we belief that this part will be useful.  (Line 117-135)

The results should be compared to other approach not with random algorithm. Since the figure's resolution in Figure 2 is low, itis difficult to observe.

Answer:   Due to the research has been implement and compare result with the random algorithm therefore, we put in the future research to use another approaches to compare the result of this study.  (Line 571-574)  

We redraw the new Figure 2 to be more clear. (Line 503-508)

Round 2

Reviewer 1 Report

-

Reviewer 3 Report

The paper may have significantly improved. However, the authors have failed to significantly address most of the issues raised. For instance, the conclusion section still looks like a literature review, having some citations. This is not conventional. The conclusion section should simply underscore the major contributions of the paper, results, limitations and future research directions without having in-text citations. 

Because of my previous comment on the algorithm, the authors went to rewrite this algorithm which was wholly lifted from another paper without duly referencing this paper. This portrays unethical conduct on the part of the authors. 

Most of my previous comments were not properly addressed and the issue of plagiarism was totally ignored. 

Reviewer 4 Report

Thank you for revising the manuscript. It satisfied the reviewer's comments.